First case of a reversed Parabothus taiwanensis Amaoka & Shen, 1993 from Taiwan (Pleuronectiformes: Bothidae), with first evidence of situs inversus viscerum in bothid

Su Yo 1 2
Ho Hsuan-Ching ogcoho@gmail.com 2 3 4
Chu Tah-Wei 2
1 Department of Marine Biotechnology and Resources, National Sun Yat-sen University , Kaohsiung , Taiwan
2 Department and Graduate Institute of Aquaculture, National Kaohsiung University of Science and Technology , Kaohsiung , Taiwan
3 Institute of Marine Biology, National Donghwa University , Pingtung , Taiwan
4 Australian Museum , Sydney , Australia
Nganvongpanit Korakot
Electronic publication date: 2024 Feb 23
Publication date: 2024
Volume: 12
Electronic Location ID: e16829
Received 2023 Oct 6; Accepted 2024 Jan 4
Copyright: ©2024 Su et al.
Copyright year: 2024
Copyright holder: Su et al.
License: This is an open access article distributed under the terms of the Creative Commons Attribution License, which permits unrestricted use, distribution, reproduction and adaptation in any medium and for any purpose provided that it is properly attributed. For attribution, the original author(s), title, publication source (PeerJ) and either DOI or URL of the article must be cited.
License URL: https://creativecommons.org/licenses/by/4.0/

Keywords: Abnormality, Morphology, Reversed eye, Flatfish, Ichthyology, Dextral

Funding: National Kaohsiung University of Science and Technology, Kaohsiung, Taiwan This study was supported by National Kaohsiung University of Science and Technology, Kaohsiung, Taiwan. The funders had no role in study design, data collection and analysis, decision to publish, or preparation of the manuscript.

==============================
Background

Reversed condition is rarely found in most flatfishes in natural environment, except for some certain species. The mechanism controlling the reversals in flatfishes has been studied in some cultivated species, whereas some have only few cases for the entire family and remain unclear. Here, we report the first record of a dextral (reversed) specimen of Parabothus taiwanensis Amaoka & Shen, 1993 collected off southwestern Taiwan recently. It represents the second reversed case ever recorded in Bothidae. We aim to provide a detailed description of this dextral specimen and compared to the sinistral (normal) specimens collected from the vicinity.

Methods

Specimens were fixed in 4% formaldehyde and transferred to 70% ethanol for preservation. Meristic and morphometric characters were examined for both dextral and sinistral specimens. Dissections were made on specimens to confirm the position of internal organs. Lastly, X-radiographs were taken to elucidate the osteological features.

Results

As a result, no differences of both meristic and morphometric characters were observed between the dextral and sinistral specimens. Nevertheless, situs inversus viscerum is discovered in the dextral specimen for the first time in Bothidae and the sixth record within flatfishes.

Introduction

The sinistral flatfish family Bothidae, currently comprised of 20 genera and 169 species, are widely distributed in tropical to temperate waters globally (Amaoka & Ho, 2019; Fricke, Eschmeyer & van der Laan, 2023). Among them, the genus Parabothus (Norman, 1931) can be discriminated from other bothid genera by having an elliptical body; tip of isthmus below or slightly behind the posterior margin of the eye; scales on ocular side with moderately long ctenii; eyes separated by a concave space; usually absence of rostral spine in males; no white blotches on anterior margin of head (Amaoka, Mihara & Rivaton, 1997; Amaoka & Ho, 2019).

There are currently ten species recognized as valid and distributed in the Indo-West Pacific Ocean (Voronina, Pruvost & Causse, 2017), with four species currently recognized in Taiwanese waters, namely, Parabothus coarctatus (Gilbert, 1905), Parabothus kiensis (Tanaka, 1918), Parabothus polylepis (Alcock, 1889), and Parabothus taiwanensis Amaoka & Shen, 1993 (Amaoka & Ho, 2019). Among these species, P. taiwanensis was originally described from Taiwan based on five types collected off Kaohsiung, southwestern Taiwan. The species is common in Taiwan and widespread in the western Pacific Ocean including Japan, Ryuku Islands, Taiwan, and Vanuatu (Nakabo & Doiuchi, 2013; Voronina & Causse, 2014; Amaoka, 2019).

Larval flatfishes resemble typical fishes which have a bilaterally symmetrical body form and swim in an upright position in the water (Schreiber, 2006). These larvae undergo metamorphosis into juvenile (immature adult) form. During this process, one of the eyes migrates to the opposite side of the head and transforms into a lateralized swimming posture (Schreiber, 2006). Some flatfish species, for example, Paralichthys lethostigma and Psettodes erumei, can metamorphose into either sinistral or dextral morphology and behavior (Hensley, 2001; Schreiber, 2013). Nonetheless, among approximately 715 known species of flatfishes, only seven exhibit variations in dextral to sinistral polymorphism within geographically distinct populations, while the rest appear to be consistent on one side. Moreover, the exact selective advantage of sinistral versus dextral form still remains unclear (Schreiber, 2013).

In most flatfish families, the reversal of eyes in flatfishes is considered as an abnormality, with the eyes on the opposite side of typical conspecific individuals (Diaz de Astarloa, 1997). In addition, the abnormal specimens exhibit several characteristics that are either reversed or abnormal. For example, the pectoral fin is longer on the ocular side; pelvic-fin origin on the ocular side anterior to that of the blind side; incomplete coloration of the blind side; retention of blind-side coloration on the ocular side; olfactory lobes and nasals are well-developed on the ocular side; situs inversus viscerum; the crossing of optic chiasma (Follett, McCormick & Best, 1960; Okiyama & Tomi, 1970; Amaoka, 1964; Moore & Posey, 1972; Ivankov, Ivankova & Vinnikov, 2008; Goto, 2009; Kuroshima, Obata & Kawai, 2022). This phenomenon has been occasionally observed in several flatfish families, for instance: Trinectes maculatus (Achiridae); Citharoides macrolepidotus (Citharidae); Symphurus plagiusa (Cynoglossidae); Citharichthys spilopterus and Etropus cyclosquamus (Paralichthyidae); Cleisthenes pinetorum, Glyptocephalus stelleri, and Microstomus achne (Pleuronectidae); and Poecilopsetta plinthus (Poecilopsettidae) (Hubbs, 1945; Amaoka, 1964; Moore & Posey, 1972; Ruiz-Carus & Rider, 1998; Ivankov, Ivankova & Vinnikov, 2008; Goto, 2009). In some cases of Paralichthys flounders, the vast majority of wild-caught specimens are sinistral; however, when cultured in a laboratory or aquacultural environment, the incidence of offspring with “reversed” (dextral) asymmetry has been reported to be much higher than those found in nature (Schreiber, 2013).

Accordingly, the case of reversed form appeared to be extremely rare in Bothidae. Only Kuroshima, Obata & Kawai (2022) reported a reversal case of Chascanopsetta lugubris based on a single specimen, representing the first record, and also the only one in Bothidae. Moreover, they found no osteological differences between the dextral and sinistral specimens.

Previously, situs inversus viscerum was only found in five species of flatfishes, i.e., Symphurus plagiusa (Cynoglossidae), Glyptocephalus zachirus and Tanakius kitaharae (Paralichthyidae); Microstomus achne and Cleisthenes pinetorum (Pleuronectidae) (Hubbs & Hubbs, 1944; Follett, McCormick & Best, 1960; Dahlberg, 1970; Goto, 2009; Table 1), and it has never been recorded in bothids. Moreover, such condition was not observed in the first reserved bothid case of Chascanopsetta lugubris (i.e., internal organs not reversed) (Kuroshima, Obata & Kawai, 2022).

Table 1 List of all records of reversed flatfish with situs inversus viscerum.

Family	Species	n	Reference	Situs inversus viscerum	Crossing of optic chiasma	
Bothidae	Parabothus taiwanensis	1	This study	+	N/A	
Cynoglossidae	Symphurus plagiusa	1	Dahlberg (1970)	+	N/A	
Pleuronectidae	Cleisthenes pinetorum	1	Goto (2009)	+	N/A	
Microstomus achne	3	+	N/A	
Glyptocephalus zachirus	2	Follett, McCormick & Best (1960)	+	+	
Tanakius kitaharae	1	Hubbs & Hubbs (1944)	+	+	
Notes.

N/A, not available.

Recently, an unusual dextral flatfish specimen was collected from the Ke-Tzu-Liao fishing port, southwestern Taiwan. After a close examination, this specimen is confirmed as a reversed form of Parabothus taiwanensis Amaoka & Shen, 1993. Examination of the same species deposited in the collection reveals that it is the only reversed specimen found in this species, and also the second specimen in the family. Moreover, the situs inversus viscerum is also discovered in our specimen which represents the first case of situs inversus viscerum in the Bothidae, and the sixth record in flatfishes. Here we describe the dextral specimen, compare it to conspecific sinistral specimens, and document the first finding of situs inversus viscerum within Bothidae in history.

Materials & Methods

Methods for taking counts and measurements follow Hubbs & Lagler (1974) and Amaoka, Mihara & Rivaton (1993). Numbers of caudal-fin rays are expressed as upper unbranched ray + branched ray + lower unbranched ray. Gill rakers were counted on the outer face of the first ocular-side gill arch, with the raker at angle included in the lower-raker count. Scale pockets were adopted when scales were missing due to bottom-trawl operation. The vertebral formulae were determined by a digital X-ray machine (Dexela CMOS X-ray detector, Model 2315, Dexeal Co. Ltd., UK) set up in the National Museum of Marine Biology and Aquarium, Pingtung, Taiwan. Measurements were made using digital calipers rounding to the nearest 0.1 mm. Data of morphometric characters were presented as percentages or ratios of standard length (SL) and head length (HL), except where otherwise indicated. Terminology of osteology follow Amaoka (1969). The distribution map was generated from Ocean Data View (Schlitzer, 2023).

Specimens are deposited in the Pisces Collection of the National Museum of Marine Biology and Aquarium , Taiwan (NMMB-P). Counts and measurements of 13 specimens of P. taiwanensis, including one dextral and 12 sinistral specimens were taken. The dextral specimen (NMMB-P 38889) and one sinistral specimen (NMMB-P 38874) were dissected to examine the internal organs. Determination of sex follow Amaoka & Ho (2019), with specimens possess rostral spine, mandibular knob, and a rather wide interorbital space identified as male, others as female or juvenile.

Ethical statements

All materials used were either museum collections or collected by us from local fish markets and were dead when obtained. No living animals were used in this study.

Results

Family Bothidae	
Parabothus taiwanensis Amaoka & Shen, 1993	
Figures 1 & 2, 4 and 5. Tables 2 and 3.	

Parabothus taiwanensis Amaoka & Shen, 1993:1042 (Type locality: Kaohsiung, Taiwan. Holotype: HUMZ 114127. Paratypes: HUMZ 114128, NTUM 05591–05592, NTUM 05599). Amaoka, Mihara & Rivaton, 1997: 161 (mentioned, compared to the new species described). Amaoka in Randall & Lim, 2000: 645 (listed). Nakabo in Nakabo, 2002: 1361 (in key, description). Ho & Shao, 2011: 61 (listed). Shen & Wu, 2012: 754 (in part, description). Nakabo & Doiuchi in Nakabo, 2013: 1665 (in key, description). Voronina & Causse, 2014: 149 (new record from Vanuatu). Voronina, Pruvost & Causse, 2017:275 (mentioned, compared to the new species described). Amaoka & Ho, 2019: 202 (in part, description). Amaoka in Koeda & Ho, 2019: 1244 (in part, description). Fricke, Golani & Appelbaum-Golani, 2019: 370 (in key).

Figure 1 Dextral specimen of Parabothus taiwanensis.Amaoka & Shen, 1993, NMMB-P38889, 95.2 mm SL.

(A) Fresh. (B) Preserved. (C) X-radiograph. Photos by Yo Su.

Figure 2 Illustration of the position of visceral organs of Parabothus taiwanensisAmaoka & Shen, 1993, NMMB-P38889, 95.2 mm SL.

Left: ocular side; right: blind side. Figure not scale. Illustrated by Yo Su.

Parabothus chlorospilus (non Gilbert): (Shen, 1983):13 (in part, misidentification).

Parabothus sp.: (Shen, 1983): 13 (in part, description). Shen in Shen, 1993: 571 (in part, description).

Crossorhombus sp.: (Shen, 1983): 19 (in part, description).

Specimen examined

Dextral: NMMB-P 38889 (1 male, 95.2 mm SL), off Ke-Tzu-Liao fishing port (ca. 22°42′53″N, 120° 13′12″E), Kaohsiung, southwestern Taiwan, 8 July 2023, coll. Y. Su.

Description of dextral specimen. Meristic and morphometric data are provided in Tables 2 and 3.

Dorsal-fin rays 103. Ocular-side pectoral-fin rays 13; blind-side pectoral-fin rays 10. Ocular-side pelvic-fin rays 6; blind-side pelvic-fin rays 6. Anal-fin rays 81. Caudal-fin rays 2 + 13 + 2. Gill rakers 0 + 8 = 8. Lateral-line scales 60, not including 3 scales on caudal fin. Vertebrae 10 + 29 = 39.

Table 2 Comparison of meristic characters between dextral and sinistral specimens of Parabothus taiwanensisAmaoka & Shen, 1993.

	Dextral NMMB-P38890	Sinistral n = 12	
Dorsal-fin rays	103	94–109	
Pectoral-fin rays (O)	13	11–13	
Pectoral-fin rays (B)	10	10–11	
Pelvic-fin rays (O)	6	6	
Pelvic-fin rays (B)	6	6	
Anal-fin rays	81	76–85	
Caudal-fin rays	2 + 13 + 2	2 + 13 + 2	
Gill rakers	0 + 8 = 8	0 + 7–8 = 7–8	
Lateral-line scales	60	57–64	
Lateral-line scales on caudal fin	3	3–4	
Vertebrae	10 + 29 = 39	10 + 29–30 = 39–40	
Notes.

Abbreviations B blind side

O ocular side

Table 3 Comparison of morphometric characters between dextral and sinistral specimens of Parabothus taiwanensisAmaoka & Shen, 1993.

	Dextral NMMB-P38890	Sinistral n = 12	
SL (mm)	95.2	74.4–129.5	
%SL		Mean (range)	SD	
HL	24.5	25.5 (24.0–27.0)	0.9	
Body depth	43.8	43.7 (40.1–46.4)	1.8	
Snout length	4.6	4.9 (4.5–5.2)	0.2	
Upper-eye diameter	8.2	8.4 (7.4–9.0)	0.5	
Lower-eye diameter	7.8	8.1 (7.3–8.9)	0.5	
Interorbital width (M)	3.0	3.3 (2.3–4.5)	0.7	
Interorbital width (F & J)	–	1.9 (1.4–2.5)	0.5	
Upper-jaw length (O)	8.5	9.0 (8.5–9.9)	0.5	
Upper-jaw length (B)	8.1	9.0 (8.2–9.8)	0.6	
Lower-jaw length (O)	11.5	12.0 (11.4–12.7)	0.4	
Lower-jaw length (B)	12.7	12.7 (11.8–13.4)	0.5	
Caudal-peduncle depth	10.7	10.4 (9.8–10.9)	0.4	
Pectoral-fin length (O)	15.9	16.7 (15.2–18.0)	0.9	
Pectoral-fin length (B)	9.3	9.2 (8.1–10.1)	0.7	
Pelvic-fin base (O)	8.0	8.4 (7.1–9.1)	0.5	
Pelvic-fin base (B)	3.9	4.2 (3.8–4.9)	0.4	
Longest dorsal-fin ray	11.7	12.7 (10.9–15.4)	1.2	
Longest anal-fin ray	13.0	12.8 (11.0–14.9)	1.1	
Caudal-fin length	20.0	20.9 (19.0–22.1)	1.0	
Notes.

Abbreviations B blind side

F female

HL head length

J juvenile

M male

O ocular side

SD standard deviation

SL standard length

Body elliptical, greatest depth 2.3 in SL; both dorsal and ventral profiles of body convex and nearly asymmetric. Head moderate, length 4.1 in SL; anterior profile of head steep, and with deep concavity in front of interorbital space. Snout short, length 5.4 in HL, distinctly shorter than eye diameter. Rostral spine present but blunt. Eyes small, upper-eye diameter 3.0 in HL, and lower-eye diameter 3.2 in SL; lower eye slightly in advance of upper eye. Orbital spines absent. Interorbital space concave, its width 8.1 in HL. Two nostrils present on ocular side and situated anterior to lower eye; anterior nostril forming tube and with small flap; posterior nostril without flap, situated immediately in front of lower eye; nostrils on blind side present, situated below dorsal-fin origin.

Mouth large, oblique; upper-jaw length on ocular side 2.9 in HL; its anterior tip at same vertical through tip of lower jaw; its posterior tip at same vertical through anterior margin of upper eye. Upper-jaw teeth on both ocular and blind sides biserial at anterior half, those on outer row rather stouter; lower-jaw teeth on both ocular and blind sides uniserial, slightly larger and more-spaced than those on upper jaw.

Gill rakers present on outer-four arches; those on first arch short, laterally compressed; first raker on lower arch with several spinules on inner face, while others sooth, not serrated; those on second and third arches about same size as those on first arch; rakers on fourth arch distinctly shorter; no rakers on upper limb of all four arches. Scales on ocular side present and deciduous; scales absent on snout and both ocular-side jaws; scales on blind side deciduous.

Dorsal-fin origin before midline of interorbital space; anterior most rays not elongated; longest ray at slightly posterior to middle of body. Anal-fin origin at same vertical through base of ocular-side pectoral fin; its outer margin subsymmetrical to dorsal-fin margin. Membranes of dorsal and anal fins perforated at base. Pectoral fins on both ocular and blind sides short, length 1.5 and 2.6 in HL, respectively. Origin of ocular-side pelvic fin at tip of isthmus; origin of blind-side pelvic fin opposite to fourth ray of ocular side. Tip of isthmus at same vertical through posterior margin of lower eye. Caudal fin slightly pointed. All rays simple except for middle 13 caudal-fin rays.

Coloration

When fresh (Fig. 1A), body light brown, with abdominal region dusky. All fin rays slightly darker than body color. When preserved (Fig. 1B), body coloration similar to fresh but slightly yellowish; all fin rays dusky; peritoneum scattered with dense black pigments.

Osteology

Neural spines absent on first vertebra; parapophysis well-developed on fifth to tenth vertebra; pleural ribs and epipleurals absent (Fig. 1C). Four plates on caudal skeleton, including three hypurals and parhypural; epural fused with second uroneural and sixth hypural; urostyle fused with fourth to fifth hypurals; second and third hypurals fused. Distal margin of all hypural plates unbranched and no distinct clefts or grooves.

Visceral organs

Anus situated on blind side and immediately anterior to first anal-fin ray; genital papilla situated on ocular side, slightly anterior to anus (Fig. 2). Liver situated on ocular side of abdominal cavity; stomach and intestine coils on blind side of abdominal cavity.

Distribution

The distribution map is shown in Fig. 3. This species is widely distributed in the western Pacific Ocean, including Japan (Nakabo & Doiuchi, 2013), East China Sea (Yamada et al., 2007), Taiwan (Amaoka & Ho, 2019), and Vanuatu (Voronina & Causse, 2014). Inhabits depths usually less than 100 m in Taiwan (Amaoka & Ho, 2019); down to 249–252 m in Vanuatu (Voronina & Causse, 2014).

Figure 3 Distribution of Parabothus taiwanensis in Taiwanese waters (shaded in black), and the location of Ke-Tzu-Liao fishing port (red star).

The information of distribution was based on records from Yamada et al. (2007), Nakabo & Doiuchi (2013), and Amaoka & Ho (2019). The map was generated from Ocean Data View (Schlitzer, 2023; https://odv.awi.de/).

Remarks

The present specimen is identified as Parabothus taiwanensis (Amaoka & Shen, 1993) in having 103 dorsal-fin rays; 81 anal-fin rays; 60 lateral-line scales; a short and blunt rostral spine present in male; biserial upper-jaw teeth; and perforations present on dorsal and anal fins (Voronina & Causse, 2014; Amaoka & Ho, 2019). Moreover, the meristic and morphometric characters of the dextral specimen fall in the range of other sinistral specimens (Tables 2 and 3).

Despite the position of the eyes, our dextral specimen possesses several characteristics that are also reversed compared to normal specimens (Figs. 4 and 5): (1) length of the right (ocular) side pelvic-fin base longer than the left (blind) side, (2) length of the right-side pectoral fin longer than the left side, (3) left eye slightly posterior to the right eye, (4) lateral line well-developed on the right side, (5) a blunt rostral spine on the right side, (6) anus on the left side, (7) genital papilla on the right side, (8) liver on right side of the abdominal cavity, and (9) stomach and intestine coils on left side of the abdominal cavity (Fig. 2).

Figure 4 Sinistral specimens of Parabothus taiwanensisAmaoka & Shen, 1993.

(A) NMMB-P 38874, male, 96.9 mm SL, fresh condition. (B) NMMB-P 24854, one or two specimens, female, 122.0 mm SL, preserved condition. Photos by Yo Su.

Figure 5 X-radiographs of sinistral specimens of Parabothus taiwanensis (Amaoka & Shen, 1993).

(A) NMMB-P 23272, 1 of 2 specimens, female, 129.5 mm SL. (B) NMMB-P 38874, male, 96.9 mm SL. Photos by Yo Su.

Discussion

Several cases (mainly Paralichthys spp.) showed that there are higher incidences of reversal in cultured flounders than in the wild populations, and the reasons for this phenomenon still remain unknown (Schreiber, 2013). However, it was suggested several environmental parameters may affect the selection and frequency of sinistral and dextral forms (Schreiber, 2013). With the extremely rare cases (only 2 specimens) in bothid species, it is difficult to explain whether the reversed forms are naturally rare or have been selected in the natural environment.

The first case of situs inversus viscerum in Bothidae

Situs inversus viscerum was only found in a few flatfish species and this was not found in the first reversed bothid, Chascanopsetta lugubris, i.e., internal organs not reversed, according to Kuroshima, Obata & Kawai (2022). In contrast, situs inversus viscerum is present in our specimen, which is the first case in the family Bothidae. In normal individuals, the liver is on the left side of the abdominal cavity, whereas the stomach and intestine coils are on the right side (Amaoka, 1964), however, the reversed specimen has its liver, stomach, and intestine coils on the opposite side of sinistral specimens.

Hashimoto et al. (2002) isolated a reversed clonal line of homozygous Japanese flounder (Paralichthys olivaceus) through gynogenesis, and found that the reversed individuals have higher frequency of the reversal of visceral organs than normal individuals. Moreover, they found that the reversal of the visceral organs was not always accompanied by the reversal of eye migration during metamorphosis. Therefore, they proposed that the orientation of eye migration is not correlated with the structural or molecular data associated with the L/R axis in the visceral organs, and different mechanisms should be involved in controlling the reversed locus in the Japanese flounder.

Based on the aforementioned results, we may conclude that although the situs inversus viscerum may have a higher chance in reversed flatfishes. However, the mechanism still remains unknown, and it appears to be a somewhat random condition among flatfishes.

Conclusions

In this study, we reported the first dextral Parabothus taiwanensis from Taiwan, representing the second specimen ever recorded within Bothidae. Both meristic and morphometric characters show no difference between the dextral and sinistral specimens. Additionally, situs inversus viscerum was observed in the dextral specimen, which is the first record in Bothidae and the sixth record among flatfishes.

Comparative materials

Sinistral specimens of Parabothus taiwanensis: NMMB-P 23219 (two juvenile, 74.4–85.4 mm SL), off Ke-Tzu-Liao fishing port, 30 April. 2015, coll. H.-C. Ho. NMMB-P 23232 (three male, one female, 92.7–110.2), off Ke-Tzu-Liao fishing port, 31 March 2016, coll. H.-C. Ho. NMMB-P 23272 (one male, one female, 115.8–129.5), off Ke-Tzu-Liao fishing port, 28 February 2016, coll. H.-C. Ho. NMMB-P 24854 (one male, one female, 115.8–122.0), off Ke-Tzu-Liao fishing port, 2 April 2015, coll. H.-C. Ho. NMMB-P 25695, (one juvenile, 74.9), off Ke-Tzu-Liao fishing port, 27 June 2016. NMMB-P 38874 (one male, 96.9), off Ke-Tzu-Liao fishing port, 25 June 2023, coll. Y. Su.

Supplemental Information

Supplemental Information 1 Raw data of counts and measurements of specimens

We thank H. C. Lin (NSYSU) for providing resource materials and access to facilities; K. Amaoka (Hokkaido University) for providing literature and critical comments; Tung-Hai Fishery Company for assistance in sample collection; P. N. Lee (NMMBA) for curatorial assistance and specimen loan; M. H. Jiang (NMMBA) for assistance in taking X-radiographs.

Additional Information and Declarations

Competing Interests

Author Contributions

Data Availability

The authors declare there are no competing interests.

Yo Su conceived and designed the experiments, performed the experiments, analyzed the data, prepared figures and/or tables, authored or reviewed drafts of the article, and approved the final draft.

Hsuan-Ching Ho performed the experiments, analyzed the data, prepared figures and/or tables, authored or reviewed drafts of the article, and approved the final draft.

Tah-Wei Chu analyzed the data, authored or reviewed drafts of the article, and approved the final draft.

The following information was supplied regarding data availability:

The raw measurements are available in the Supplementary File.

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
