# Peer review of "First case of a reversed Parabothus taiwanensis Amaoka & Shen, 1993 from Taiwan (Pleuronectiformes: Bothidae), with first evidence of situs inversus viscerum in bothid"

_PeerJ, doi:10.7717/peerj.16829_

## Round 0.1 · original submission · Major Revisions

Please revise this manuscript following the comments from the editor and reviewers point by point and highlight all text changes in this revised manuscript. In your rebuttal letter, you must address all comments from the editor and the reviewers and refer to the main text.

**Language Note:** The review process has identified that the English language must be improved. PeerJ can provide language editing services - please contact us at copyediting@peerj.com for pricing (be sure to provide your manuscript number and title). Alternatively, you should make your own arrangements to improve the language quality and provide details in your response letter. – PeerJ Staff

Reviewer 1 ·

Basic reporting

Overall, I found that this manuscript is well-written and provided sufficient background knowledge of the sinistral flatfish from family Bothidae. Also, finding of new specimen as the first case of dextral Parabothus taiwanensis and first case of situs inversus viscerum is an interesting topic. Nevertheless, my main concern is in the methods and discussion section that are certainly needed to be revised. I also have some suggestions and comments in the general comments section below for the improvement of this manuscript.

Experimental design

In the introduction section, the author had mentioned about the examination of the other sinistral specimens comparing with the dextral specimens (Line 93) in this study. Also, the meristic and morphometric characters of sinistral (n=12) is appeared in the Table 2 &3 and also figures 3 & 4. However, there is lack of the description in the materials and methods section for the measurement of these 12 sinistral specimens or what the author did with these specimens. I suggest providing more detail about the other used 12 Sinistral specimens in the materials and methods section.

Also, the sex of this dextral specimen had been mentioned as a male, but the author did not provide the method for sex determination. I suggest describing more in detail.

Validity of the findings

I consider this study is important to reveal the hidden variation of the morphology of this flatfish in the family Bothidae. However, the materials and methods should be improved as the suggestions I provided.

Additional comments

The keywords should not repeat the word existed in the title. I suggest replacing the repeating words by new words for better chance of being discorded the article, for example, flatfish, dextral and sinistral.

Line 47-48: The author mentioned about the four species currently recognized in Taiwanese water but did not give the species name. Please, provide the list of these four species in the manuscript.

Line 53-60: These lines need the references.

Line 62-63: The reversal of eyes in flatfishes is considered an abnormality, with the eyes on the opposite side of typical conspecific individuals (REF).

Line 112-113: In the result section, please add the subsection “taxonomic review” before the beginning of family Bothidae description.

Line 141: In this part, the author provided the morphometric characters in the proportional data without any previous mention. Then, I suggest that the author should provide more information, perhaps in the materials and methods section as you used the proportional measurement in SL and HL.

For discussion section, it will be better if the author could relocate the text from line 192-204 to the results section as there is only the explanation of what the author found in this study (repeating result), but for line 205-210 are the examples of a good discussion. I suggest starting the discussion with the highlight of the study then the authors could list the other interesting topics that have been made from the introduction section. For example, what are the possibly reasons of raring dextral form for Parabothus taiwanensis in the natural habitat? Or is it possible to find the other dextral Parabothus taiwanensis in other areas and why?

Line 216: Please remove “(Fig. 2”) from the line as it is not necessary to be repeated in this section.

Figures and Tables
The figures and the description of the figures are not matched (Fig1 ,2). Please, recheck and correct it.

I understand that Figures 3 and 4 of the sinistral specimens were provide for the comparison between normal and dextral, but these figures also need to be mentioned in the text. Please do the same with Tables 2 and 3, you provided information of sinistral specimen, but the comparison should be provided in the text as I mentioned above.

In addition, I suggest rearranging the figure number for “Figs 3 & 4” which are the sinistral (normal) Parabothus taiwanensis should be stated as the first order (Change them to Figure 1 and Figure 2 instead) because you described the normal Parabothus taiwanensis first in the result section. Then, Figs 1 & 2, the dextral specimen should be in the later order (Change them to Figure 3 and Figure 4 instead). Then, Figs 3 & 4 (rearranged) can be mentioned on line 133, 169 and 176. Also, do not forget to rename the figures for the rest of manuscript.

Reviewer 2 ·

Basic reporting

When I was reading the manuscript with the tittle of First case of a reversed Parabothus taiwanensis Amaoka & Shen, 1993 from Taiwan (Pleuronectiformes: Bothidae), with first record of situs inversus viscerum in bothid (#91443), I just followed the chartered structure of Parabothus taiwanensis using several basic-methods.

However, the weakness of this paper including the description of the study in all parts. Several minor/majors are noted as the yellow highlight.

Experimental design

Introduction
In the introduction, the author should be proposed the important for this study, but it is still missing (see the attached file).

Methodology
In the methodology, it did not mention in detail where the method was developed from the standard method.

Use of laboratory animals should be subjected to some guidelines set out by the ethics committee. Indicate how the fish caught were sacrificed and the guidelines used in the action.
(other comments see the attached file)

Results
The results of the study did not explain, once the method has been developed. It would be clearer if all figures are labeled in relation to the results/content. The way of presenting the results makes it difficult to locate the structures anatomically without the labeling data.

In many moments the description and the photos including Fig. 1C and Figs 4 (bone and cartilage staining method) do not allow to visualize the described information. The color figures are showed.
(other comments see the attached file)


Discussion
Since no labels of the figure results are found, the discussion of this manuscript is very limited.
Some data of the function and important evolution of Situs inversus viscerum are unclear in the discussion. It is not happy when I was reading and think “how and why” but the discussion is not clear.
I suggest some improvements, the work could focus on showing specific characters of the group and make a discussion focused on the group's information. What can Situs inversus viscerum anatomy infer about the adaptive evolution of this group of fish that occupies a very atypical habitat within the entire fish Class.

Validity of the findings

-

Additional comments

-

Annotated reviews are not available for download in order to protect the identity of reviewers who chose to remain anonymous.

Reviewer 3 ·

Basic reporting

This study would like to present the first case of a reversed Parabothus taiwanensis (dextral) in family Bothidae. The backgrounds of flounders about the eyes that migrates to the opposite side in flatfish. The figures given were high quality, well labelled and described. However, I have some suggest improving the current ms.

Experimental design

no comment

Validity of the findings

no comment

Additional comments

1. The English language should be improved a bit. An example where the language could be edited include line 62-the current sentence seemed incorrect.
2. Please give approval for animal ethics at where to be mentioned.
3. The number of specimens used in this study should be provided in materials and methods.
4. The X-ray machine should be cited for the brand and company and country.
5. In Table 1, the meaning or fullname of N/A should be described in footnote.
6. Table 3 seems to be weird for the row of SL and %SL. I suggest revising it. The full name of SL and HL should be mentioned in footnote.
7. I suggest you show the locality where collected the sample in map in the ms.

The ms is written with clear context with professional language although there is a few to be revied. I recommend the current ms should be improved upon before Acceptance.

---

## Round 0.2 · Minor Revisions

Thank you very much for your revised manuscript. The reviewers have provided a few comments that require your response before I can make a decision.

Reviewer 1 ·

Basic reporting

I am satisfied with the revised version of the manuscript, and I saw a lot of improvements here; however, there is some minor points that still need to be revised and rearranged.

Experimental design

-

Validity of the findings

-

Additional comments

The figures should be mentioned in the manuscript as the numerical order. Please relocate the lines from 197-200 (Coloration) to the line before Osteology (line 185), as the figures can be arranged in numerical order (Fig. 1A, 1B, 1C then Fig. 2 Visceral organs).

Also, please relocate (Fig. 1C) in line 185 into the text line.
Example,

Osteology
Neural spines absent on first vertebra; parapophysis well-developed on fifth to tenth vertebra; pleural ribs and epipleurals absent (Fig. 1C). Four plates on caudal skeleton, including three hypurals and parhypural; epural fused with second uroneural and sixth hypural; urostyle fused with fourth to fifth hypurals; second and third hypurals fused. Distal margin of all hypural plates unbranched and no distinct clefts or grooves.

Please do the same with (Fig. 2) in the line 192.

Reviewer 2 ·

Basic reporting

Nice

Experimental design

Several responses from the author are nice, however, I think that the dead specimens or carcass fishes should be processed via the animal used committees.

Please indicate the approval no. given by the university ethics committee for animal study and also attach a scanned copy of the approval statement.

Question: Use of laboratory animals should be subjected to some guidelines set out by the ethics committee. Indicate how the fish caught were sacrificed and the guidelines used in the action.
(other comments see the attached file)
Answer: The specimens used in this study were either long-preserved museum specimens or collected by us in the local fish market and were already dead when the fishing boat arrived at the harbor, therefore no living or laboratory animals were used. This is stated at Line 123-125.

Validity of the findings

Nice

Additional comments

None

Reviewer 3 ·

Basic reporting

no comment

Experimental design

no comment

Validity of the findings

no comment

Additional comments

I believe that the revised ms is suitable for publication.

---

## Round 0.3 · accepted · Accept

Congratulations on your manuscript; it is ready to be published.